# The Neural Network Prior Kernel

## Abstract

Deep neural networks have been shown to incorporate certain inductive biases and structural information about the input data, even at initialization when the network weights are randomly sampled from a prior distribution. We show that this phenomenon may be linked to a new type of neural network kernel, which we call the Neural Network Prior Kernel (NNPK). The NNPK value between two input examples is given by the expected inner product of their logits, where the expectation is calculated with respect to the prior weight distribution of the network. Although the NNPK is inherently infinite-dimensional, we study its properties empirically via a finite-sample approximation by representing the input examples as vectors of finitely many logits obtained from repeated random initializations of a network. Our analysis suggests that certain structures in the data that emerge from a trained model are already present at initialization. Our findings also indicate that the NNPK conveys information about the generalization performance of architectures. We then provide a theoretical result that connects the NNPK to the Neural Tangent Kernel (NTK) for infinitely-wide networks. We validate this connection empirically for a number of standard image classification models. Finally, we present an application of the NNPK for dataset distillation based on kernel ridge regression.

## 1 Introduction: Neural Kernels

There has been tremendous progress in recent years in connecting deep neural networks with kernel machines. The main example of such efforts is the NTK (Jacot et al., 2018) which introduces a kernel in terms of the gradient of the network weights with respect to the inputs. the NTK has provided significant insight into the training dynamics (Canziani et al., 2016; Novak et al., 2018a; Li & Liang, 2018; Du et al., 2019; Allen-Zhu et al., 2019) and generalization properties (Arora et al., 2019a; Cao & Gu, 2019) of neural networks. The properties of the NTK are well studied in the infinite-width regime and under a squared loss assumption (Lee et al., 2019; Arora et al., 2019b). Recent progress has generalized these results to hinge loss and support vector machines (Chen et al., 2021). Also, recently, it was shown that the training dynamics of any network trained with gradient descent induces a path kernel that depends on the NTK (Domingos, 2020).

Another class of neural models involves constructing limit cases of infinitely wide Bayesian neural networks, resulting in a Gaussian process (NNGP) (Lee et al., 2017; Matthews et al., 2018; Novak et al., 2018b; Garriga-Alonso et al., 2018; Borovykh, 2018). The NNGP describes the distribution of the output predictions of a randomly initialized infinitely-wide network. Adopting a sequential view of neural networks casts the uncertainty in the output of a layer as a conditional distribution on the previous layer's output. The NNGP is closely related to the NTK (Jacot et al., 2018; Hron et al., 2020) and has been used to characterize the trainability of different architectures (Schoenholz et al., 2016). We discuss their distinctions in Section 2.2.

Neural networks have been shown to have great expressive power even when the weights are randomly initialized (Frankle & Carbin, 2018; Evci et al., 2020; Ramanujan et al., 2020). For instance, only training the BatchNorm (Ioffe & Szegedy, 2015) variables in a randomly initialized ResNet model provide a reasonable test performance on CIFAR and ImageNet-1k datasets (Frankle et al., 2020). In our work, we extract features from a repeatedly randomly re-initialized neural network. We show that these random features are, in fact, associated with a new kernel.

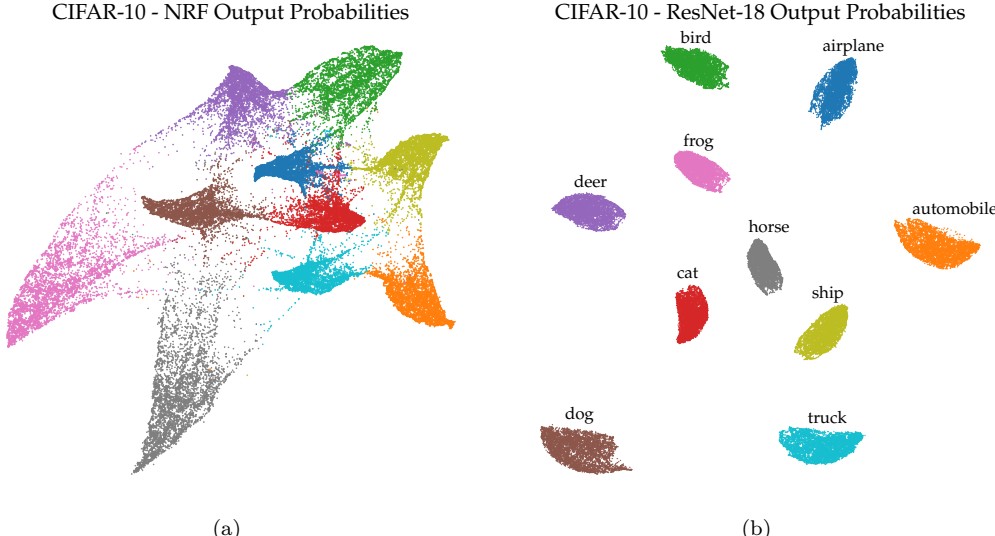

Figure 1: TriMap (Amid & Warmuth, 2019) visualization of the CIFAR-10 training examples: (a) output probabilities of a (linear) logistic classifier trained on a 10,240-dimensional NNPK representation, extracted from identical *randomly initialized* ResNet-18 architectures, (b) output probabilities of the fully trained model. Note that the clusters in (a) already appear in (b), and the closeness of pairs of clusters is roughly preserved.

Our new kernel is called the Neural Network Prior Kernel (NNPK). For two input examples, it is defined as the expected value of the inner product of the logits of the network with respect to the random initialization of the weights.[1] As this expectation is an integral over all possible realizations of the prior distribution on the network's weights, the NNPK inherently corresponds to an infinite-dimensional representation in a Hilbert space. Our finite-sample approximation of the NNPK consists of representing the input examples as a vector of logit features that are computed based on a sample of random initializations of the network drawn from the prior. We show in extensive experiments with the NNPK across different architectures, initialization priors, activation functions, and dimensions suggest that the NNPK is a useful tool to analyze the trainability of neural networks. Also, our experiments reveal that certain structures in the data that manifest in a fully trained neural network are already present at initialization. Thus, the NNPK may also provide further insights into the expressive power of neural networks. We also give a theoretical result that connects the NNPK to the NTK for infinitely-wide networks. We evaluate this result empirically on a number of standard image classification architectures. Finally, we provide an application of the NNPK for dataset distillation using Kernel Inducing Points (KIP) (Nguyen et al., 2021).

Figure 1 shows a TriMap (Amid & Warmuth, 2019) visualization of the output probabilities of the training examples of CIFAR-10 obtained from a linear classifier that is trained on 10,240 sampled logit features extracted from *randomly initialized* ResNet-18 models. This is contrasted with the output probabilities obtained from the same ResNet-18 model after being trained on the original dataset. The visualizations reveal a fair amount of similarity between the structure of the data and the placement of the clusters induced by the randomly initialized network features and the fully trained network. This suggests that the final learned structure of the data is highly correlated with the initial projected structure, perceived by the randomly initialized network.

## 2 The Neural Network Prior Kernel

### 2.1 Definition

We now formally define the Neural Network Prior Kernel (NNPK) of a neural network. Let $f_{\boldsymbol{w}}^k : \mathbb{R}^d \to \mathbb{R}^k$ denote a neural network function parameterized by the weights $\boldsymbol{w} \in \mathbb{R}^p$. The network transforms an input

---

[1]Similarly to NNGP, we construct the kernel by correlating the output predictions; however, NNGP concerns an infinitely wide randomly initialized network, while we consider infinite ensemble of finite-width networks with popular architectures.

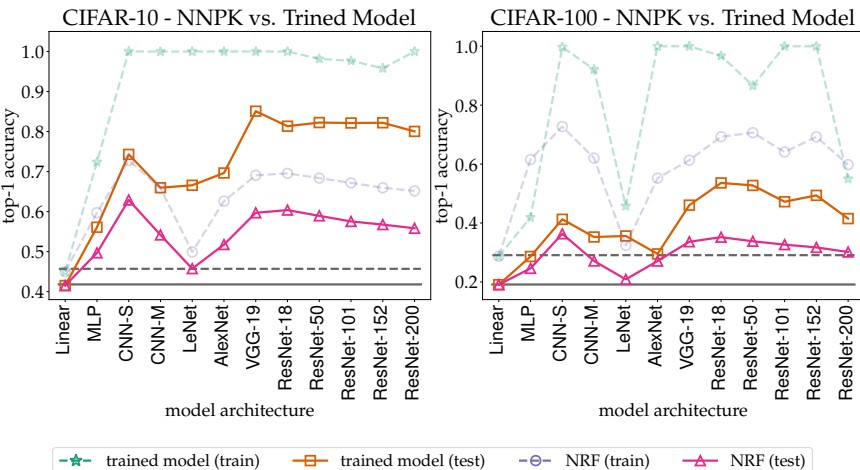

Figure 2: The NNPK performance of different model architectures on CIFAR-10/100 datasets: Train and test accuracy of linear classifiers trained on an NNPK approximation via randomly sampled $n = 3072$ features from different standard architectures. The accuracy of the trained networks on the original CIFAR-10/100 train data is also shown for each network. The models are sorted roughly in the order of their capacity. We also plot a simple linear classifier's train and test accuracy on the original inputs using dashed and solid horizontal lines, respectively. Note that the train and test curves for learning with the NNPK present rough approximations of the corresponding curves for the fully trained models.

$\boldsymbol{x} \in \mathbb{R}^d$ to the $k$-dimensional output $f_{\boldsymbol{w}}^k \in \mathbb{R}^k$, called *logits*. The logits are the output of a linear fully-connected layer and are fed into a softmax function to produce a $k$-dimensional probability vector over classes. The weights $\boldsymbol{w}$ at the initialization follow a *prior* distribution[2] $\pi(\boldsymbol{w})$. For a deep neural network, this distribution usually amounts to a scaled (truncated) Normal or a uniform distribution. The NNPK of the network between two inputs $\boldsymbol{x}, \boldsymbol{x}' \in \mathbb{R}^d$ is then defined in terms of an expectation with respect to the initial weight distribution $\pi$,

$$\kappa_{\text{NNPK}}(\boldsymbol{x}, \boldsymbol{x}') = \mathbb{E}_{\boldsymbol{w} \sim \pi}[\langle f_{\boldsymbol{w}}^k(\boldsymbol{x}), f_{\boldsymbol{w}}^k(\boldsymbol{x}') \rangle]. \tag{NNPK}$$

That is, the NNPK is simply the expected value of the inner product of the output logits for the given two inputs. Clearly, the NNPK is a valid kernel since it corresponds to a convex combination of symmetric and positive semi-definitive functions. The NNPK is also deterministic for a given distribution $\pi$, as the expectation is over all realizations of $\boldsymbol{w} \sim \pi$.

We can calculate a finite sample approximation of the NNPK as

$$\kappa_{\text{NNPK}}^{(n)}(\boldsymbol{x}, \boldsymbol{x}') = \frac{1}{n} \sum_i \langle f_{\boldsymbol{w}_i}^k(\boldsymbol{x}), f_{\boldsymbol{w}_i}^k(\boldsymbol{x}') \rangle, \tag{1}$$

using $n$ iid samples $\{\boldsymbol{w}_i\}_{i=1}^n$ drawn from $\pi$. For a given neural network architecture, this corresponds to randomly re-initializing the networks $n$ times and calculating the average inner product between the logits for the given two inputs.

For a randomly initialized network, the output logits of an example are correlated. In order to obtain uncorrelated output samples, in our construction, we set $k = 1$ while keeping the rest of the network architecture unchanged. In this case, each randomly initialized network maps each input example to a single scalar value. Thus, Eq. (1) can be written as

$$\kappa_{\text{NNPK}}^{(n)}(\boldsymbol{x}, \boldsymbol{x}') = \langle \boldsymbol{\phi}_n(\boldsymbol{x}), \boldsymbol{\phi}_n(\boldsymbol{x}') \rangle, \tag{2}$$

---

[2]The term *prior* is commonly used in Bayesian statistics to describe the uncertainty of a distribution before observing some evidence. Here, we loosely use the term prior to denote the initial distribution of the model weights, although the connection to Bayesian statistics is yet to be established in future work.

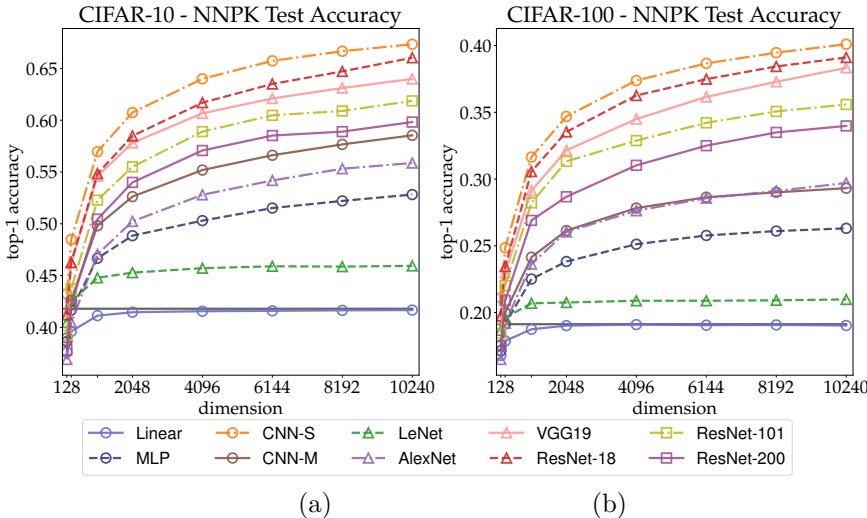

(a)            (b)

Figure 3: Effect of dimension $n$ on the finite-sample approximation of the NNPK using $\phi_n$: Test accuracy of linear classifiers on (a) CIFAR-10 and (b) CIFAR-100, trained on $\phi_n$ with varying dimensionality $n$ extracted from different model architectures. We also plot the test accuracy of the linear model trained on the original CIFAR-10/100 input images (solid horizontal lines). As we expect, the accuracy of the linear random projection of the input images (Linear) does not exceed the accuracy on the original inputs, as we increase the dimension. On the contrary, the accuracy of the classifiers trained on $\phi_n$ improves monotonically with dimension. This justifies the definition of the NNPK as a limit case when $n \to \infty$. Also, the order of the performance using different model architectures remains roughly the same across dimension.

where the embedding map $\phi_n : \mathbb{R}^d \to \mathbb{R}^n$ is defined as

$$\phi_n(\boldsymbol{x}) = {}^1\!/\!\sqrt{n} \left[ f^1_{\boldsymbol{w}_1}(\boldsymbol{x}), \ldots, f^1_{\boldsymbol{w}_n}(\boldsymbol{x}) \right]^\top . \tag{3}$$

These randomly generated features are then treated as new representations for the training examples and are used to train a classifier. In our experiments, we mainly focus on training a linear classifier with a softmax output to predict the class probabilities. At inference, the same randomly initialized networks are used to construct the random features for a test example. Note that the NNPK approximation is different from random feature construction for approximating a kernel proposed by Rahimi & Recht (2007) in that we are not approximating a kernel function with a *known* closed-form. The randomness in our construction merely stems from the expectation in the definition of the kernel.

In the following sections, we show that the NNPK is a non-trivial kernel which allows separating the data significantly better than the original input space. We analyze the dependency of the NNPK (via its finite-sample approximation) on different elements and properties of the models. Interestingly, the NNPK is a fixed kernel for a given architecture, and therefore, the same randomly initialized networks to approximate the NNPK can be used across different datasets. This is in contrast to the *data-dependent* path kernel (Domingos, 2020), which relies on the training trajectory of the model.

## 2.2 Distinction from NNGP

The Neural Network Gaussian Process (NNGP) refers to the behavior of infinitely wide neural networks, as the number of hidden units goes to infinity. Specifically, NNGP is defined as the limit of the covariance matrix of the output of an infinitely wide neural network with random Gaussian weights. While NNGP and NNPK share a common definition in terms of the inner product of the logits, the main distinction between the two is that NNGP is defined for infinitely wide neural networks, while NNPK applies to networks of any width. Additionally, unlike NNPK, which involves an expectation over all realizations of the network weights at initialization, computing NNGP does not involve any expectation. Thus, while approximating the NNGP is a computationally involved procedure for finite-width networks (Novak et al., 2022), NNPK can be approximated efficiently via Eq. (3).

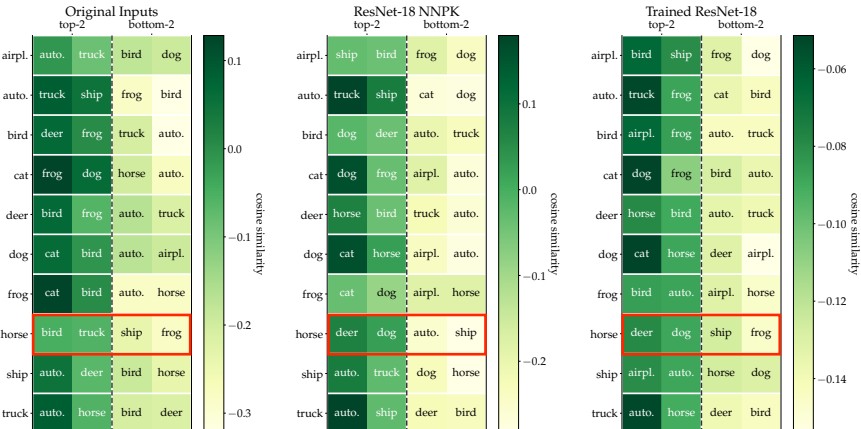

Figure 4: Cosine similarities for different classes of ResNet-18: a linear classifier trained on the original inputs (left), on the NNPK embedding sampled from ResNet-18 (middle), and the softmax layer of a fully trained ResNet-18 network (right). The NNPK appears to be an intermediate representation between the original input data and the representation induced by a fully trained model. Some classes that are close in the fully trained model are closer in the NNPK representation compared to the original input representation.

## 3 An Empirical Analysis of the NNPK

We conduct several ablations to understand the effect of different elements, such as architecture and the number of dimensions $n$ on the NNPK. For the first set of experiments, we conduct ablations on CIFAR-10/100 datasets of images (Krizhevsky, 2009). To show the generality of the features extracted from the NNPK, we also perform an experiment on the ImageNet-1K dataset (Deng et al., 2009). We defer additional results on the effect of width, depth, activation function, etc., to Appendix B.

### 3.1 Effect of Architecture

We first show the dependency of the NNPK on different model architectures. For this, we consider a number of different models of varying sizes. These models include a two-layer fully-connected network (MLP), a small (CNN-S) and a medium (CNN-M) convolutional network, variants of LeNet (LeCun et al., 1989), AlexNet (Krizhevsky et al., 2012), and VGG-19 (Simonyan & Zisserman, 2014) as well as several ResNets (He et al., 2016) (ResNet-18, ResNet-50, ResNet-101, ResNet-152, and ResNet-200). The details of all models are described in the appendix. We also consider a linear projection of the input data using a random matrix. The Linear baseline corresponds to a random projection of the input data, which is used for applications such as k-neareast neighbor search (Kleinberg, 1997) and random projection trees (Dasgupta & Freund, 2008). We use a he normal initialization (He et al., 2015) for the ResNet models and use glorot normal (Glorot & Bengio, 2010) for the rest.

We set the approximation dimension $n$ equal to the original dimensionality of the input data, which is $32 \times 32 \times 3 = 3072$. This way, any improvement in separability over the original input data is solely due to a better representation by the NNPK. To classify the data, we train a logistic regression classifier on $\phi_n$ using the L-BFGS (Liu & Nocedal, 1989) optimizer for which we tune the $L_2$-regularizer value. We repeat each experiment over 5 random trials.

In Figure 2, we plot the train and test accuracy of the NNPK along with the baseline linear classifier trained on the original input data (dashed and solid horizontal lines, respectively). We also plot the train and test accuracy of the trained network. To train each model on the training examples, we use an SGD with Nesterov momentum optimizer (Nesterov, 1983) with a batch size of 128 for 100k iterations. We use a linear warmup followed by a linear decay schedule for the learning rate and tune the maximum learning rate value, momentum constant, and the weight decay parameter for 128 trials using the Vizier Bayesian parameter search package (Song et al., 2022; Golovin et al., 2017). One important detail to notice is that we train the networks only on the original train examples *without data augmentation.* Data augmentation is standard practice for training such networks and is a strong form of regularization that allows larger networks to

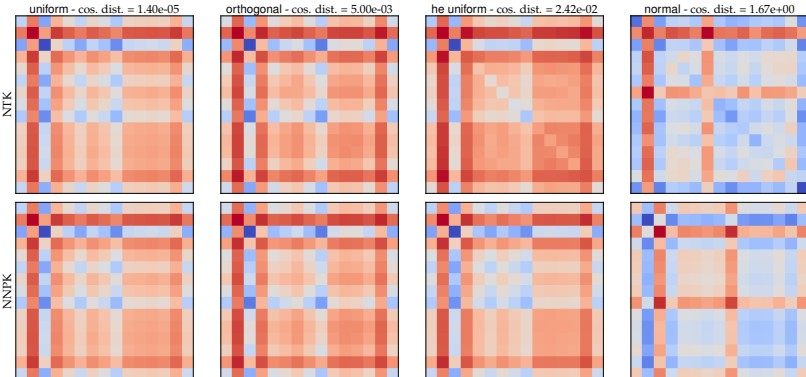

Figure 5: Comparison of the NTK and NNPK kernel matrices between 15 randomly selected CelebA examples for different prior distributions for a `ResNet-50` model. The value of the cosine distance between the full NTK and NNPK matrices over the entire batch of 1000 examples is also shown on top for each figure. The two matrices are most similar using `uniform` and are least similar using `normal`.

provide better generalization when the number of train examples is limited. With data augmentation, we expect the performance of the models to increase with size, which is evident in several existing benchmarking efforts on CIFAR-10/100 datasets.[3] However, our goal here is to understand how much a trained network can improve on the initial representation of the data, reflected in the NNPK, *after* seeing the (original) training examples.

Several interesting observations can be made from Figure 2. First, the performance of `Linear` random projection is about the same as the original data, as we expect such projections to be, at best, as good as the original data (Dasgupta & Gupta, 2003). However, the NNPK representations extracted from all architectures perform better than the baseline classifier trained on the original input image representations. Next, the performance of the NNPK is not monotonic with the architecture size. For a relatively shallower model, the performance increases up to `CNN-S`, but drops when making the model deeper, i.e., going from `CNN-S` to `LeNet` and `AlexNet`. The performance improves one more time when using the `VGG-19` architecture; however, dropping slightly but monotonically when using deeper ResNet architectures. More interestingly, the performance of the trained model also follows a similar pattern to the NNPK on both datasets.

These observations suggest that the architecture's performance on the original data without strong regularization, such as data augmentation, is highly correlated with its initial representation of the data, reflected by the NNPK at initialization. Thus, after observing the inputs, the final performance is proportional to the initial NNPK representation. We test this hypothesis further in our later experiments. Note that the initial representation tends to be more descriptive than just a random projection of the data, thus, indicating inductive biases of deep neural architectures towards better representations of natural data at initialization (Ortiz-Jimenez et al., 2020; Ortiz-Jiménez et al., 2021).

## 3.2 Effect of Dimension

We empirically show that the finite-sample approximation of the NNPK using $\phi_n$ in Eq. (3) improves for all networks as we increase the number of samples, i.e., dimension $n$. Figure 3 shows the test accuracy of linear classifiers trained on $\phi_n$ using different dimensions $n$ on CIFAR-10/100 datasets. As we expect, the performance of the random projection of the data (`Linear`) matches the performance of the linear classifier trained on the original input features (solid horizontal lines) and does not improve with the dimension. On the contrary, the performance of $\phi_n$ increases monotonically with dimension for all networks. This justifies the definition of the NNPK as a limit case when $n \to \infty$. Interestingly, the order of performance for different models (along the vertical axis) remains roughly the same across different dimensions. This suggests that the observations about the NNPK are relatively consistent using different dimensionality $n$ to approximate the kernel.

---

[3]For instance, see `https://github.com/kuangliu/pytorch-cifar` and `https://paperswithcode.com/sota/image-classification-on-cifar-10` for some benchmarking results.

### 3.3 The NNPK Reveals Structure

In this section, we show that the structure in the data that appears in a fully-trained network is somehow reflected in the NNPK. In other words, the NNPK seems to be an intermediate representation between the original input space and the final trained network. For this, we consider the linear classifier trained on the original input images from the CIFAR-10 dataset. We also consider the fully trained ResNet-18 model (without data augmentation) as well as the linear classifier trained on $\phi_n$, sampled from the randomly initialized ResNet-18 models. We set $n = 3072$ the same as the original data dimension. We then calculate the cosine similarities of the class weights for each classifier (for the fully trained ResNet-18, this corresponds to the weights of the output softmax layer). We then plot the top and bottom-2 similar classes for each of the 10 classes of CIFAR-10 in Figure 4.

We can see from Figure 4 that in many cases (such as the class 'horse'), the top similar classes are identical for the NNPK classifier and the output layer of the fully trained network. However, in some cases (e.g., 'automobile'), the top classes of the classifier trained on the original CIFAR-10 images are closer to the NNPK classifier than the ResNet-18 model. This observation further hints at the following *conjecture*: **The original representation of the data is first projected via the NNPK into an initial representation which is then enhanced throughout the training into the final representation of the fully-trained network.**

A further theoretical and empirical study of the above conjecture is an interesting future research direction.

### 3.4 Results on the Larger ImageNet-1K Dataset

In order to demonstrate the applicability of the results using the NNPK on a larger scale, we perform an experiment on the ImageNet-1K dataset (Deng et al., 2009). The dataset contains around 1.2M training images from 1000 classes. A linear classifier on the original input images (150K features) achieves a 3.4% test top-1 accuracy. In contrast, a linear classifier trained on a 4096 dimensional NNPK embedding, sampled from randomly initialized ResNet-18 models, achieves a 10.3% test accuracy. This result is fascinating given that the NNPK achieves a significantly higher accuracy with $37\times$ fewer features than the original input features. Training a linear classifier and a two-layer MLP on a 31,568 dimensional NNPK embedding achieves 12.2% and 15.2% top-1 test accuracies, respectively.

## 4 Relation to the NTK

We first provide a theoretical result that connects the NNPK to the NTK for the infinite-width regime. Let $\boldsymbol{w}_0$ denote the model weights at initialization. For any $\boldsymbol{w} \in \mathbb{R}^p$ s.t. $\|\boldsymbol{w} - \boldsymbol{w}_0\|$ is small,[4] $f_{\boldsymbol{w}}$ at $\boldsymbol{w}$ is well approximated by the linear model at initialization,[5]

$$f_{\boldsymbol{w}}(\boldsymbol{x}) \approx f_{\boldsymbol{w}_0}(\boldsymbol{x}) + \nabla f_{\boldsymbol{w}_0}(\boldsymbol{x})^\top (\boldsymbol{w} - \boldsymbol{w}_0) \,, \tag{4}$$

where $\boldsymbol{w}_0$ denotes the initial weights and $\nabla f_{\boldsymbol{w}_0}(\boldsymbol{x}) := \frac{\partial f_{\boldsymbol{w}}(\boldsymbol{x})}{\partial \boldsymbol{w}}|_{\boldsymbol{w}=\boldsymbol{w}_0}$ denotes the gradient of the network for an input $\boldsymbol{x}$ at $\boldsymbol{w}_0$. The NTK between two inputs $\boldsymbol{x}, \boldsymbol{x}' \in \mathbb{R}^d$ is defined as

$$\kappa_{\mathrm{NTK}}(\boldsymbol{x}, \boldsymbol{x}') = \phi_{\mathrm{NTK}}(\boldsymbol{x})^\top \phi_{\mathrm{NTK}}(\boldsymbol{x}') \,,$$
$$\text{where} \quad \phi_{\mathrm{NTK}}(\boldsymbol{x}) := \nabla f_{\boldsymbol{w}_0}(\boldsymbol{x}) \in \mathbb{R}^p \,. \tag{5}$$

When $f_{\boldsymbol{w}}$ in Eq. (3) is linearly approximated by Eq. (4), we can write the NNPK embedding as

$$\phi_n(\boldsymbol{x}) = c(\boldsymbol{x}, \boldsymbol{w}_0) \, \mathbf{1}_n + \frac{1}{\sqrt{n}} \, \nabla f_{\boldsymbol{w}_0}(\boldsymbol{x})^\top \boldsymbol{W} \,, \tag{6}$$

where $c(\boldsymbol{x}, \boldsymbol{w}_0) := f_{\boldsymbol{w}_0}(\boldsymbol{x}) - \nabla f_{\boldsymbol{w}_0}(\boldsymbol{x})^\top \boldsymbol{w}_0$ and

$$\boldsymbol{W} := {}^1\!/\!{}_{\sqrt{n}} \left[ \boldsymbol{w}_1, \cdots, \boldsymbol{w}_n \right] \,, \tag{7}$$

---

[4]This generally holds when $\boldsymbol{w}$ and $\boldsymbol{w}_0$ are two random initializations of the network.
[5]For simplicity, we consider a single output.

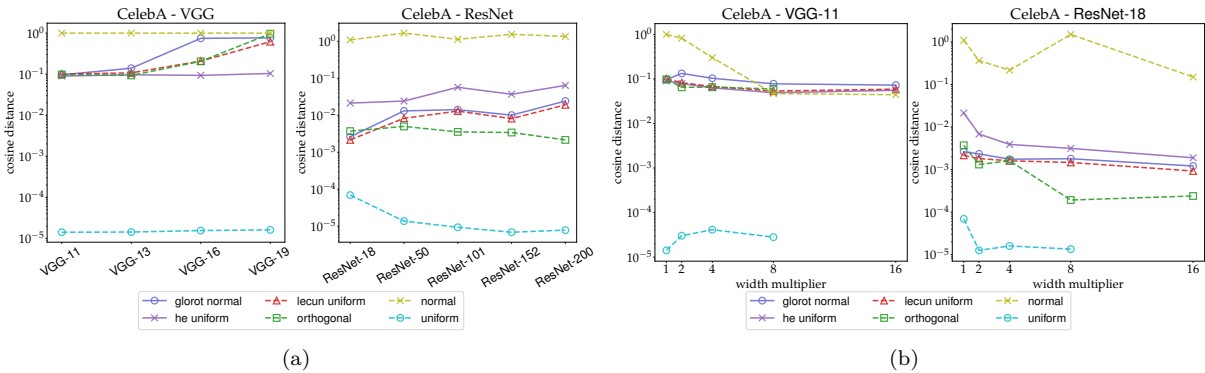

Figure 6: Effect of (a) depth and (b) width on the alignment between the NNPK and the NTK: Average cosine distance between the NNPK and the NTK kernel matrices calculated over five subsets of 1000 CelebA examples. The two kernel matrices are aligned the most (i.e., lower cosine distance) for uniform initialization and least aligned (i.e., higher cosine distance) when using normal. Also, for most initializations, the alignment decreases with increased depth and width.

in which $\boldsymbol{w}_i \in \mathbb{R}^p$, $i \in [n]$ are sampled from the network's initial (i.e., *prior*) distribution $\pi$. We now further simply the expression in Eq. (6). Firstly, note that in the linear regime,

$$c(\boldsymbol{x}, \boldsymbol{w}_0) = f_{\boldsymbol{0}_p}(\boldsymbol{x}).$$

In other words, $c(\boldsymbol{x}, \boldsymbol{w}_0)$ corresponds to the output logit of the network when all the network weights are set to zero. In practical networks, the logit is obtained by applying a linear (fully-connected) layer to the penultimate representations. Consequently, for such networks, we have $c(\boldsymbol{x}, \boldsymbol{w}_0) = 0 \,, \forall \boldsymbol{w}_0 \in \mathbb{R}^p$. Secondly, in the "lazy" regime Lee et al. (2019); Lewkowycz et al. (2020); Novak et al. (2020), which corresponds to an infinite-width network, the choice of $\boldsymbol{w}_0$ is completely arbitrary because $\forall \boldsymbol{x} \in \mathbb{R}^d$,

$$\nabla f_{\boldsymbol{w}}(\boldsymbol{x}) \equiv \nabla f_{\boldsymbol{w}'}(\boldsymbol{x}) \,, \;\; \forall \boldsymbol{w}, \boldsymbol{w}' \in \mathbb{R}^p \,. \tag{8}$$

Thus, using Eq. (5), we can write

$$\boldsymbol{\phi}_n(\boldsymbol{x}) = \boldsymbol{W}^\top \boldsymbol{\phi}_{\text{NTK}}(\boldsymbol{x}) \,. \tag{9}$$

Equation Eq. (9) shows that the NNPK embedding (via $\boldsymbol{\phi}_n$) is simply an affine transformation of the NTK embedding by the weight matrix $\boldsymbol{W}$. Moreover, we have

$$\boldsymbol{\phi}_n(\boldsymbol{x})^\top \boldsymbol{\phi}_n(\boldsymbol{x}') = \boldsymbol{\phi}_{\text{NTK}}(\boldsymbol{x})^\top \boldsymbol{W} \boldsymbol{W}^\top \boldsymbol{\phi}_{\text{NTK}}(\boldsymbol{x}') \,, \tag{10}$$

Finally, using the NNPK definition, in the limit $n \to \infty$, we can write

$$\kappa_{\text{NNPK}}(\boldsymbol{x}, \boldsymbol{x}') = \boldsymbol{\phi}_{\text{NTK}}(\boldsymbol{x})^\top \boldsymbol{\Sigma} \, \boldsymbol{\phi}_{\text{NTK}}(\boldsymbol{x}') \,. \tag{11}$$

where $\boldsymbol{\Sigma} \coloneqq \mathbb{E}_{\boldsymbol{w} \sim \pi}[\boldsymbol{w} \boldsymbol{w}^\top]$ is the covaraince matrix of the network weights. For an initialization with a zero mean and a constant scale across layers (e.g., unit Gaussian), we have $\boldsymbol{\Sigma} = \sigma^2 \boldsymbol{I}_p$ in which $\sigma^2$ denotes the variance. By applying the result to Eq. (11), we have the equivalence relation between the NNPK and the NTK at initialization as

$$\kappa_{\text{NNPK}}(\boldsymbol{x}, \boldsymbol{x}') = \sigma^2 \, \kappa_{\text{NTK}}(\boldsymbol{x}, \boldsymbol{x}') \,. \tag{12}$$

## 5  Empirical Comparison to the NTK

Although the equivalence relation between the NNPK and the NTK only holds in the infinite-width regime and for isotropic initialization, we evaluate it empirically for a number of standard image classification models. For the following experiments, we consider the classes of VGG (VGG-11, VGG-13, VGG-16, VGG-19) and ResNet (ResNet-18, ResNet-50, ResNet-101, ResNet-152, ResNet-200) models, and their variations having different width

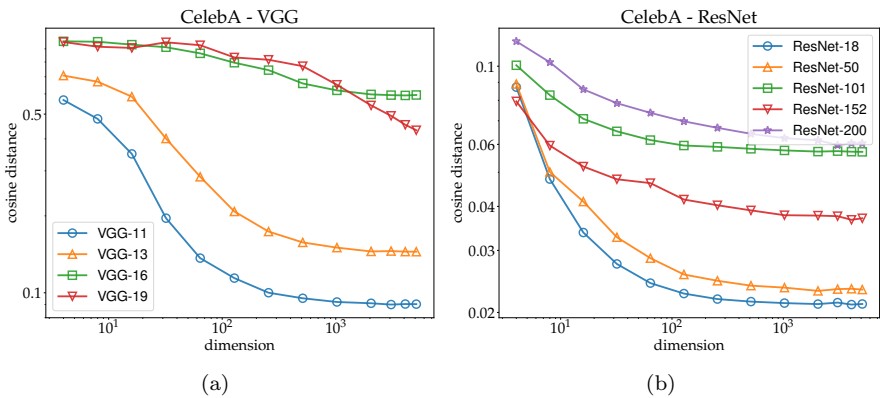

Figure 7: Effect of the NNPK dimension $n$ on the alignment with the NTK: Cosine distance between the NTK and NNPK matrices as a function of $n$ for a subset of 1000 CelebA examples using the (a) VGG-11 and (b) ResNet-18 architectures. The cosine distance decreases (i.e., higher alignment) monotonically as $n$ increases.

multipliers. We show the results on the CelebA dataset facial images dataset (Liu et al., 2015). We provide further results on CIFAR-10 and ImageNet-v2[6] datasets in Appendix C.

In each experiment, we construct the NNPK and the NTK matrices over five different subsets of 1000 examples of the dataset. For the depth and width experiments, we approximate the NNPK using $n = 1024$. We calculate the *centered* cosine distance between the two matrices, where we first subtract the mean value over all components from each matrix before calculating the standard cosine distance. The reason for centering the matrices before calculating the cosine distance is to avoid possibly large constant terms in the matrices dominating the final result. Notably, one can instead think of the alignment of the two matrices in terms of the cosine similarity, which is defined as $(1 - \text{cosine distance})$.

## 5.1 Effect of architecture and Embedding Dimension

We investigate the effect of network depth and width on the cosine distance between the NNPK and the NTK matrices. We first consider the effect of depth in Figure 4 for various VGG and ResNet architectures. The cosine distance for different architectures mostly depends on the type of initializations. We illustrate examples of the NNPK and the NTK matrices for a ResNet-50 architecture on a subset of 15 examples in Figure 5. Among these initializations, uniform induces the lowest ($\mathcal{O}(10^{-5})$) and normal yields the highest ($\mathcal{O}(1)$) distance between the kernel matrices. These initializations have a constant variance across layers (i.e., their covariance matrix corresponds to a (scaled) identity matrix). Thus, the result in Eq. (12) holds in the infinite-width case. However, in a finite width regime, only uniform initialization induces highly aligned (i.e., low cosine distance) kernels, while other properties of the network, not explained by our theory, cause normal to result in low alignment. Finally, in most cases, the distance increases with the depth of the network.

Next, we consider the VGG-11 and ResNet-18 architectures with various width multipliers for each layer. We show the average cosine distance for the same subset of examples in Figure 4.[7] Again, the initialization has the most prominent factor affecting the alignment. However, the distance between the kernels decreases for wider networks. This is consistent with our analysis which only holds for the infinitely wide case.

Lastly, we evaluate the effect of the embedding dimension $n$ on the alignment of the NNPK and the NTK. We choose a glorot normal and he uniform, respectively, for the VGG and ResNet architectures and calculate the cosine distance by varying $n$ in Figure 7. We observe that the cosine distance decreases monotonically with $n$, confirming once more that $n \to \infty$ corresponds to a valid kernel asymptotically.

---

[6]Available at `https://github.com/modestyachts/ImageNetV2`
[7]The missing values are due to the instability in the NTK package (Novak et al., 2020) for the wider variants.

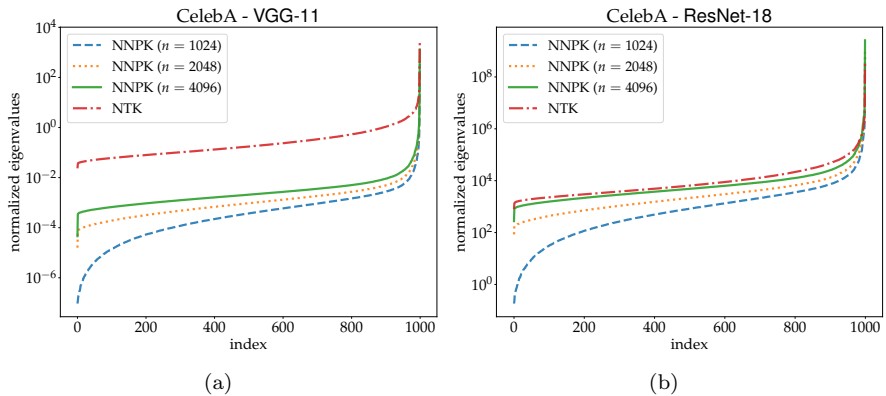

Figure 8: Comparison of the eigenspectrum of the NTK and the NNPK matrices: Normalized eignevalues (by the maximum eigenvalue) of the NTK and the NNPK matrices for a subset of 1000 CelebA examples using the (a) VGG-11 and (b) ResNet-18 architectures. As $n$ increases, the eigenspecta of the two matrices become closer.

## 5.2 Comparing the Eigenspectrum of the Kernels

Although the theoretical development in Section 4 only holds in the infinite-width regime, the NNPK and the NTK matrices still show a high level of alignment in terms of cosine similarity for standard networks and initializations. We investigate the correspondence of another property of the two kernels, namely, the eigenspectrum of the matrices. We calculate the eigenspectrum of the NTK and the NNPK matrices on 1000 CelebA examples for various values of $n$ for the VGG-11 and ResNet-18 architectures in Figure 8. We normalize the eigenspectrum of each matrix by its maximum eigenvalue. We observe from the results that the NNPK matrix induces a larger eigengap than the NTK matrix in all cases. However, the two eigenspectra become more aligned as the dimension $n$ increases. Also, the two eigenspectra are closer for the ResNet-18 model than the VGG-11.

## 6 Application: Dataset Distillation

Lastly, we consider an recent innovative data distillation method via Kernel Inducing Points (KIP) (Nguyen et al., 2021) based on Kernel Ridge Regression: A small support set $(\mathcal{X}_s, \boldsymbol{y}_s)$ is learned by minimizing the following loss on a larger data set $(\mathcal{X}_t, \boldsymbol{y}_t)$:

$$\nicefrac{1}{2}||\boldsymbol{y}_t - \boldsymbol{K}_{\mathcal{X}_t, \mathcal{X}_s}(\boldsymbol{K}_{\mathcal{X}_s, \mathcal{X}_s} + \lambda \boldsymbol{I})^{-1}\boldsymbol{y}_s||^2,$$

where for given (ordered) sets $\mathcal{U}, \mathcal{V}$, $\boldsymbol{K}_{\mathcal{U}, \mathcal{V}}$ is the matrix of kernel elements $(\boldsymbol{K}(\boldsymbol{u}, \boldsymbol{v}))_{\boldsymbol{u} \in \mathcal{U}, \boldsymbol{v} \in \mathcal{V}}$ and $\lambda > 0$ is a regularization parameter. We apply KIP to learn 10 images, one per class, from the CIFAR-10 dataset. We compare the results on a number of different kernels that are all built from a four-layer convolutional neural network having three convolutional layers with the number of filters equal to 16, followed by a fully connected dense layer.

In addition to the NNPK, we consider the NTK, which was originally proposed by Nguyen et al. (2021) for the KIP method, as well as an RBF kernel on the input images and the NNPK with $n = 1024$. We initialize the distilled images by sampling from a random Normal distribution and update them using an Adam optimizer Kingma & Ba (2014) for 2000 iterations. The results are shown in Figure 9. In the first row, we illustrate 10 sample images from the dataset for reference (which are **not** used for initialization of the distilled images). We also provide the final top-1 test accuracy on the test set is shown in Table 1. Curiously, the NNPK provides a higher test accuracy compared to the other two kernels.

Also, notice that from the second row of Figure 9, which represents the NTK results, the distilled images correspond to a rough combination of all training images. For instance, the distilled image for the class "horse" resembles a double-sided horse image. Thus, in a sense, the distilled images reveal a lot amount of information about the images in the training set, making them susceptible to backdoor privacy attacks (Liu

Table 1: Top-1 test accuracy of the KIP method on the CIFAR-10 dataset using different kernel functions.

| Kernel Type | NTK | RBF | NNPK |
|---|---|---|---|
| Top-1 Test Accuracy | 37.4 | 35.2 | **42.8** |

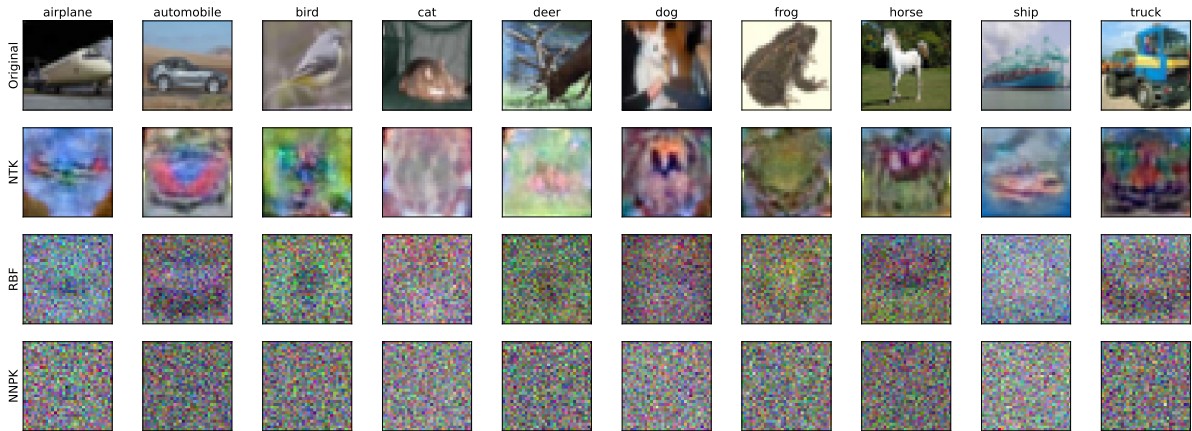

Figure 9: Distilled images via KIP using different kernel functions: (first row) original images from the CIFAR-10 train set, provided for reference, and distilled images using (second row) the NTK, (third row) RBF kernel, and (fourth row) the NNPK. Among the three approaches, the NNPK results reveal the least amount of information about the training data.

et al., 2023). Interestingly, the distilled images using RBF kernel on the third row also reveal a very similar structure to the NTK in terms of color and patterns, but in a much rougher manner. On the other hand, the NNPK results reveal even less structure about the training data. Especially, the test performance is highly dependent on the seeds for repeated initialization of the networks; with a different set of random seeds for test examples, the test accuracy drops to 12.6%. Thus, the NNPK seems ideal for cases where privacy is a concern for deploying distilled images. A more theoretical analysis of privacy aspects of the NNPK is a future research direction.

## 7 Conclusion

We introduced a new neural network kernel linked to the weights' distribution at initialization. From experiments on a large variety of image datasets and neural network architectures, we found that the representation of a fully trained network is already *reflected* in a representation induced by the NNPK. We also showed a close relation between the NNPK and the NTK for standard neural network architectures and initializations. While the NTK involves the gradient of the network, thus, requiring backward passes, the NNPK only consists of the output of a given network, thus requiring only forward passes. This line of work prompts several natural questions:

- **How is the NNPK related to the training dynamic of a given network? Can the the NNPK be used to find suitable architectures for a given dataset rapidly?** Figures 2-3 suggest that the comparative performance on the fully trained network is reflected by the NNPK.

- **How are the NNPK and the NTK related beyond the infinite-width regime?** Our findings suggest that a connection may still hold even for standard neural network architectures. Also, the properties beyond the depth and width affecting this connection are worth investigating.

- **Which other applications can benefit from using the NNPK instead of the NTK?** The computational advantage of the NNPK over the NTK (in terms of requiring only forward passes) makes it ideal for applications where the computation is a bottleneck. Also, the level of approximation of the NNPK can be controlled in terms of the dimension $n$.

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

# A    Details of the Model Architectures

MLP consists of two fully-connected layers of size 128 with relu activations followed by the output layer.

CNN-S contains two 2-D convolutional layers of size 32 and 64, respectively, with $5 \times 5$ filters. Each convolutional layer is followed by a max-pooling layer. We then apply a dense layer of size 512 followed by the final dense layer. All the activation functions are set to relu.

CNN-M contains four $5 \times 5$ convolutional layers, for which the sizes are 32, 64, 64, and 32, respectively. The first two convolutions are each followed by a max-pooling layer. We then similarly apply a dense layer of size 512 followed by the final dense layer. All the activation functions are set to relu.

LeNet is a variation of the LeNet model described in `https://www.kaggle.com/blurredmachine/lenet-architecture-a-complete-guide` in which we replace the activation functions with relu.

AlexNet is a variation of the model desctibed in `https://www.analyticsvidhya.com/blog/2021/03/introduction-to-the-architecture-of-alexnet/` where we changed the filter sizes in the first and last convolutional layers to $3 \times 3$ and $1 \times 1$, respectively.

# B    Further Empirical Analysis of the NNPK

We provide further empirical results on the effect of different components of a network on the NNPK.

## B.1    Effect of Initialization

The expectation in the definition of the NNPK is with respect to the prior distribution (i.e., initialization) of the model weights. In order to analyze the effect of the prior distribution $\pi$, we evaluate $\phi_n$ for CNN-S, LeNet, and ResNet-18 models using different standard weight initializations. As before, we set the embedding dimension $n = 3072$. Figure 10(a) shows the train (unfilled) and test (filled markers) accuracy of the linear classifiers trained on $\phi_n$. Similarly, we show the baseline train and test accuracy (dashed and solid lines, respectively) on the original inputs.

From 10(a), we observe that the NNPK representations of some networks, e.g. CNN-S are more robust to the choice of initialization than the others. The NNPK performance of the ResNet-18 architecture, in particular, seems to vary significantly and is degraded, especially when using (glorot) uniform or glorot normal initializations. In addition, the NNPK induced by the ResNet-18 model seems to perform the best when using he normal (which is usually the default initialization for ResNets (He et al., 2015)).

## B.2    Effect of Activation Function

Apart from the prior distribution on the weights (i.e., initialization), the NNPK is also dependant on the activation functions of the network. To study their effect, we conduct an experiment by varying the activation functions of CNN-S, LeNet, and ResNet-18 models. Again, we set the embedding dimension $n = 3072$ and use the default initializations for each model.

From Figure 10(b), we can see that certain activation functions such as leaky relu (slope $= 0.1$) seem to consistently produce better NNPK representations across architectures (although elu seems to perform the best on ResNet-18, which is known to also perform well for a trained ResNet (Shah et al., 2016)). On the other hand, the NNPK representation induced by the sigmoid activation function is consistently worse for all architectures. This suggests that the choice of a good activation function has a significant effect on the initial NNPK representation of the model, thus it may as well affect the generalization performance of the model after training.

In summary, based on our experiments on different initializations and activation functions and our findings on ResNet-18, the NNPK seems to hint at the choices that are known to work also well for fully trained models (Gotmare et al., 2018; Shah et al., 2016; Martens et al., 2021).

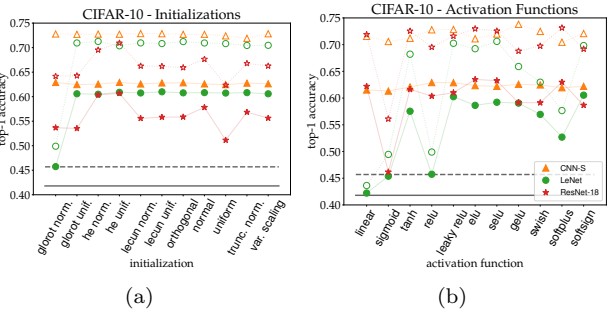 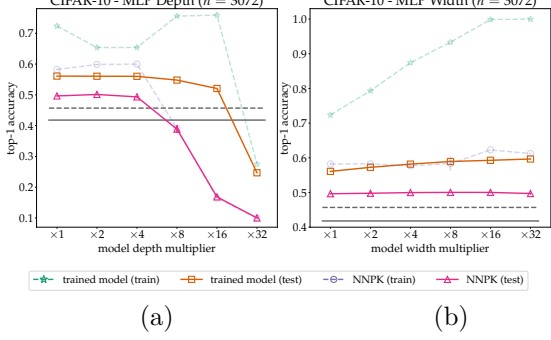

(a) (b)  (a) (b)

Figure 10: Effect of initialization and activation function on the NNPK on CIFAR-10: Train (unfilled) and test (filled markers) accuracy of linear classifiers on $\phi_n$ with $n = 3072$, sampled from CNN-S, LeNet, and ResNet-18 architectures using different (a) initializations and (b) activation functions. The train and test accuracy of a linear classifier on the original CIFAR-10 inputs are shown using dashed and solid horizontal lines, respectively. Note that for fully trained ResNet networks, he normal initialization and elu activations are known to achieve good performance (He et al., 2015; Shah et al., 2016). Surprisingly, the NNPK picks out these choices in (a) and (b).

Figure 11: Effect of depth and width on the NNPK on CIFAR-10: Train and test accuracy of linear classifiers on $\phi_n$ with $n = 3072$, extracted from an MLP model with varying (a) depth and (b) width. We also plot the train and test accuracy of the trained networks (without data augmentation). We also show the train and test accuracy of a linear classifier on the original CIFAR-10 inputs using dashed and solid horizontal lines, respectively. Note again that the NNPK performance curves are rough approximations of the curves for the fully trained models.

## B.3 Effect of Depth and Width

A natural question is whether depth or width affects the NNPK of a network. To study this, we create different variants of our basic MLP model by expanding its width or depth (i.e., number of layers) by different multiples. Figure 11 illustrates the train and test accuracy of these expanded models as a function of the multiplier. We use the default glorot normal initialization for the networks. We also plot the train and test accuracy of the trained models (using the same tuning protocol and without data augmentation). Interestingly, the NNPK does not improve by increasing the depth of the network. Also, the deeper networks become harder to train without data augmentation and other regularization techniques. In contrast, both the performance of the NNPK and the trained network improves slightly by increasing the width. This is also consistent with the empirical findings on wide neural networks (Novak et al., 2018a; Canziani et al., 2016).

## B.4 Does Skip Connection Improve the NNPK Representation?

The immense success of ResNets is mainly attributed to two elements: the use of Batch Normalization (BatchNorm) (Ioffe & Szegedy, 2015) and skip connections (He et al., 2016). There has been a number of efforts to replace or remove these two elements from ResNets (Zhang et al., 2019; Gaur et al., 2020; Bachlechner et al., 2021). The more recent adjustments proposed in (Martens et al., 2021) have shown further promise. This modification includes replacing the relu activation function with a scaled leaky relu and the use of delta orthogonal initialization for the convolutional kernels (and orthogonal initialization for the final dense layer). Here, we explore the dependence of the NNPK and the trained models (without data augmentation) on skip connections (and BatchNorm). One important point to notice is that BatchNorm is an affine transformation on the activations in each layer during inference. Since the mean and scale values are initially set to 0 and 1, respectively, BatchNorm with default initialization will have no effect on the NNPK, but affects the performance of the trained model. We also examine whether the NNPK varies by removing skip connections and whether the changes proposed in (Martens et al., 2021) affect the results.

We create variants of the ResNet-18 model by removing BatchNorm and/or skip connections. In the first approach, we use the default he normal initialization with relu activation while in the second case, we use the delta orthogonal initialization proposed in (Martens et al., 2021) for the convolutional filters (and orthogonal

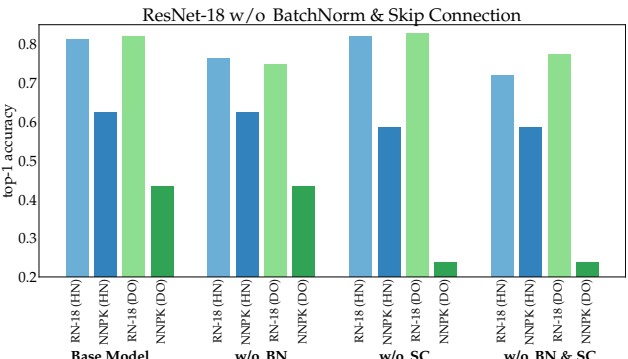

Figure 12: Effect of BatchNorm and skip connection on the ResNet-18 model and the induced NNPK represenation $\phi_n$ on CIFAR-10: we consider the original ResNet-18 model with he normal initialization and relu activations (RN-18 (HN)) along with the variant proposed in (Martens et al., 2021) with delta orthogonal initialization and scaled leaky relu activations (RN-18 (DO)). We also generate a $n = 3072$ dimensional NNPK representation using each model. We plot the test accuracy of each trained model (without data augmentation) as well as the performance of linear classifiers trained on the NNPK representation for the base models along with the variants in which BatchNorm and/or skip connections are removed. As we expect, we observe that BatchNorm with the default initial values behaves as an identity map, thus has no effect on the NNPK. On the other hand, removing skip connections degrades the performance on the NNPK.

initialization for the final dense layer). We also replace the relu activation with the scaled leaky relu proposed in (Martens et al., 2021). We set the slope of the leaky relu to 0.3. For both variants, we train the models without data augmentation and tune the learning rate, momentum constant, and the weight decay parameters for 128 trials. We generate $n = 3072$ dimensional $\phi_n$ using each variant and train linear classifiers. The results are shown in Figure 12.

It can be seen from Figure 12 that the performance of the trained base model (using he normal initialization and relu activation) degrades without BatchNorm (and skip connection). However, the performance remains about the same without only the skip connections. As we expect, the performance using the NNPK representation of the base model does not vary by adding/removing BatchNorm. However, removing skip connections degrades the performance of the NNPK.

In comparison, in most cases, the trained model using delta orthogonal initialization and leaky relu activation performs better than the base model, especially when both BatchNorm and skip connections are removed. The NNPK performance of this model also follows a similar pattern: the NNPK does not vary by adding/removing BatchNorm, but degrades without the skip connections. However, the performance of the NNPK representation of this model is comparatively lower than the one of the base model.

These observations are interesting from several aspects. First, the NNPK representation seems to improve by using skip connections for both initialization and activation function combinations. However, the performance of the trained model using the adjustments proposed in (Martens et al., 2021) seems to improve without the skip connections. This observation shows that there are cases for which the NNPK may not reflect the performance of the final trained model (although the results may vary when using data augmentation and other types of regularizations). Additionally, as we expect, the NNPK is not affected by BatchNorm using the default initialization for the BatchNorm variables. However, the dependence of the NNPK on different types of normalization techniques such as Layer Normalization (Ba et al., 2016) and Instance Normalization (Ulyanov et al., 2016) is yet to be examined.

## C    Additional Results on the Comparison to the NTK

We provide further empirical results on the relation between the NNPK and the NTK using the CIFAR-10 and ImageNet-v2 datasets. Similarly, we consider the classes of VGG (VGG-11, VGG-13, VGG-16, VGG-19) and ResNet (ResNet-18, ResNet-50, ResNet-101, ResNet-152, ResNet-200) models, and their variations having different

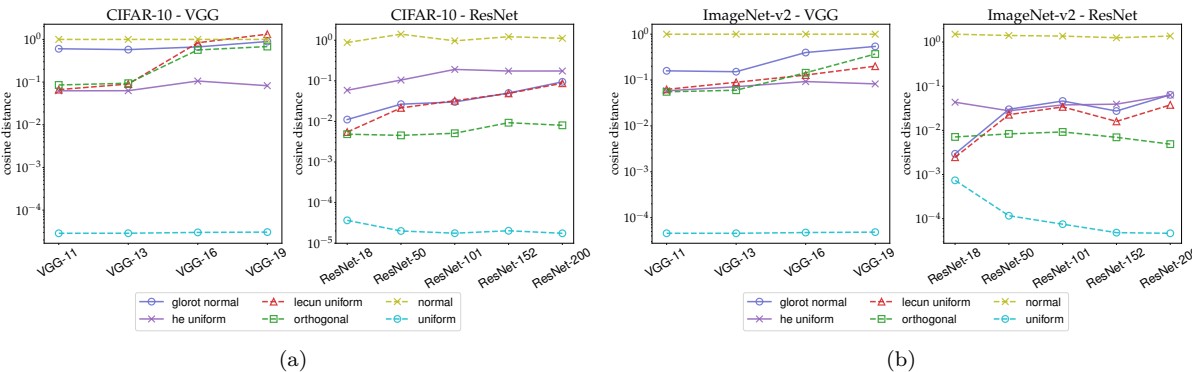

Figure 13: Effect of the NNPK depth on the alignment between the NNPK and the NTK: Cosine distance between the NTK and NNPK matrices as a function depth calculated over five subsets of 1000 examples (a) CIFAR-10 and (b) ImageNet-v2 datasets. For most initializations, the cosine distance increases (i.e., lower alignment) as the depth increases.

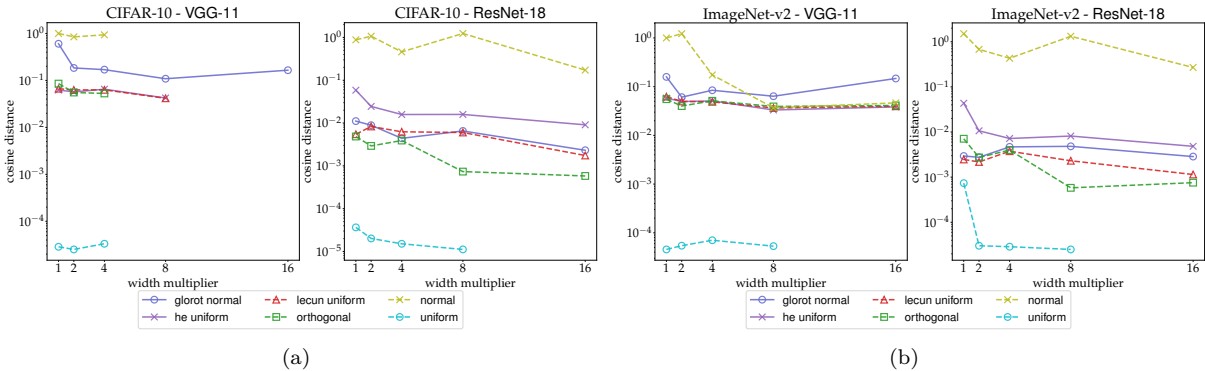

Figure 14: Effect of width on the alignment between between the NNPK and the NTK: Cosine distance between the NTK and NNPK matrices calculated over five subsets of 1000 examples as a function of the width multiplier for (a) CIFAR-10 and (b) ImageNet-v2 datasets. For most initializations, the cosine distance decreases (i.e., higher alignment) as the width increases.

width multipliers. Also, in each experiment, we construct the NNPK and NTK matrices over five different subsets of 1000 examples of the dataset. For the depth and width experiments, we approximate the NNPK using $n = 1024$.

We first show the effect of depth in Figure 13 and the effect of width in Figure 14. The conclusions are similar to the ones drawn for the CelebA dataset: the cosine distance mostly depends on the initialization, and generally increases with larger depth and decreases for wider networks.

We also illustrate the effect of the embedding dimension $n$ on the alignment of the NNPK and the NTK for the two datasets in Figure 15. The alignment between the two kernels increases with $n$ for both datasets.

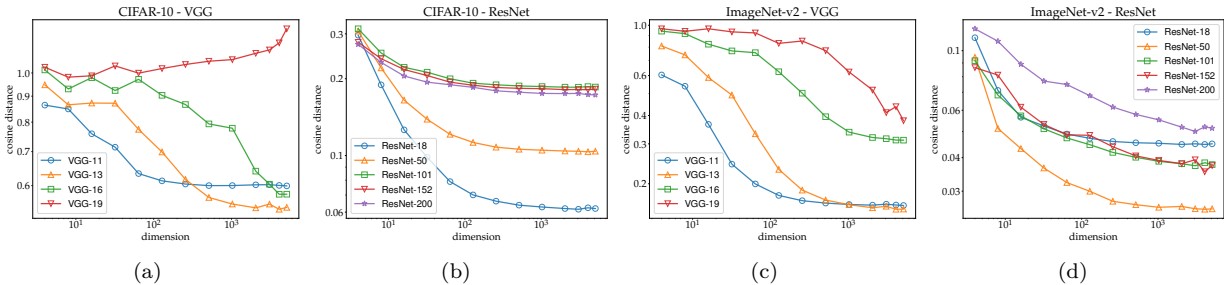

(a)              (b)             (c)             (d)

Figure 15: Effect of the NNPK dimension $n$ on the alignment between the NNPK and the NTK: Cosine distance between the NTK and NNPK matrices as a function of $n$ calculated over five subsets of 1000 examples for the (dataset, model) pairs: (a) (CIFAR-10, VGG-11), (b) (CIFAR-10, ResNet-18), (c) (ImageNet-v2, VGG-11), and (d) (ImageNet-v2, ResNet-18). The cosine distance decreases (i.e., higher alignment) monotonically as $n$ increases in general.

