# OpenReview forum: "The Neural Network Prior Kernel"
_TMLR — Withdrawn by Authors_

### Review · Reviewer_T4uF · 2023-04-21

**Summary Of Contributions:**

This work proposes a Neural Network Prior Kernel (NNPK) that associates with a random feature model where each feature is a DNN with random weights. Unlike NTK and NNGP, this kernel is defined for a finite-width NN and can be easily computed on data points. In addition, this work studies the empirical performance of NNPKs and their relation to the performance of corresponding NNs under various architectures. It also explores the theoretical and empirical relation between NNPK and NTK. Finally, the paper shows the potential application of the NNPK to dataset distillation via Kernel Inducing Points method.

**Audience:**

Yes

**Broader Impact Concerns:**

None.

**Claims And Evidence:**

No

**Requested Changes:**

Changes for acceptance:
1. [theoretical] The theoretical derivation in Section 4 may not be corrected. Alternatively, I suggest to explicitly compute NTK, NNPK and NNGP for a two-layer NN for comparison.
2. [experimental] To improve the experiments in Fig. 2, I suggest ensuring that training accuracy arrives at 100% to avoid undesired interference from insufficient training. Additionally, presenting detailed experimental setups can help convince the audience that the comparison is fair.

Changes to strength the work:
1. [clarification] In Section 1, the paper states "This suggests that the final learned structure of the data is highly correlated with the initial projected structure, perceived by the randomly initialized network." I am not sure in which sense there is high correlation between Figure 1(a) and (b) other than they are clustered.
2. [clarification] In the bottom of page 4, the paper states "while approximating the NNGP is a computationally involved procedure for finite-width networks (Novak et al., 2022)", however, I do not find similar comment about NNGP in (Novak et al., 2022).
3. [clarification] One of the main conclusions that "the representation of a fully trained network is already reflected in a representation induced by the NNPK." is too vague without a clear explanation of the exact meaning of "reflect".
4. [typo] Figure 2: "Trined Model"-> "Trained Model"

**Strengths And Weaknesses:**

Strength:
The idea of NNPK is interesting and is more relevant to finite-width NNs than NNGP.

Weakness:
1. The theoretical derivation in Section 4 regarding the equivalence between NNPK and NTK is incorrect. Linear approximation Eq. (4) at w0 well holds in a neighbourhood of w0 when the NTK is almost a constant. This in general requires ||w-w0||/||w0||<<1, not ||w-w0||<<1 as stated in the paper. As a consequence, w=0 is clearly out of the neighbourhood of any w0. All later derivation incorrectly uses the linear approximation Eq. (4), which results in the false theoretical conclusion that NNPK and NTK are equivalent.
2. The experiments in Fig. 2 showing the relevance between the NNPK performance and the NN performance are flawed. When one studies the generalization performance of NNs, the training accuracy by default should arrive at 100% to avoid undesired interference from insufficient training. However, in both CIFAR-10 and CIFAR-100, many large models clearly capable of fitting all training data do not arrive at 100% training accuracy. Therefore, the test error curves are invalid to be compared to the NNPK result.

---

### Review · Reviewer_BhqA · 2023-04-30

**Summary Of Contributions:**

The paper defines Neural Network Prior Kernel (NNPK) which is the expected value of the inner product of the transformed features at a neural net's logit layer with random weights. The paper provides numerical evidence that the defined kernel function behaves similarly to the well-known neural tangent kernel (NTK). Section 4 provides a theoretical discussion which aims to explain the connection between NNPK and NTK kernels.

**Audience:**

Yes

**Broader Impact Concerns:**

Not applicable to this paper

**Claims And Evidence:**

No

**Requested Changes:**

1) The introduction needs to clarify the motivation for NNPK kernels and answer what would be the added value of NNPK in deep learning studies.

2) Section 4 needs to be revised and the theoretical conclusions should be stated as theorems, where the assumptions behind the conclusions are clearly stated and explained.

3) Section 6 needs to explain the connection between the numerical observations and the alignment between NTK and NNPK.

4) The literature review needs to discuss the related works on kernel functions based on randomly weighted neural nets.

**Strengths And Weaknesses:**

Strength:

1) The paper proposes the concept of Neural Network Prior Kernel (NNPK) which could be useful for the theoretical and numerical analyses of deep learning problems.

Weaknesses:

1) The authors' motivation for proposing the NNPK kernel is not clear in the paper. The paper provides a definition of NNPK and tries to numerically show the connection to the well-known NTK, but does not explain how NNPK can help with the applications or theoretical analyses of neural networks. I think the paper needs to provide concrete examples of theoretical or numerical analyses of neural nets where the defined NNPK can be applied to improve the results.

2) The theoretical statements in section 4 seem to be based on strong assumptions which could highly restrict their application to general deep learning settings. Here is a list of such assumptions which seem to be used to reach the final conclusion in equations (10-12):

a. In equation (6), the authors consider the NNPK of the linear approximation of the neural net instead of the neural net itself. It seems rather expected that the NNPK of the linearly-approximated neural net function will be equivalent to the NTK kernel. This assumption will lead to ignoring all the complexity that comes from the non-linearity of the neural net function.

b. The assumption in Equation (8) seems quite strong and supposes that the gradient of the neural net does not depend on the choice of weight parameter $\mathbf{w}$. I am wondering whether the authors need this assumption to show Equations 10-12.

3) In addition to the above comments on section 4, I recommend replacing the discussion in this section with one or two theorems where the assumptions are clearly stated and discussed. The writing of section 4 needs to be improved, and it is not easy to find the assumptions behind the section's final conclusions in equations 10-12.

4) The final conclusion of section 6 sounds somewhat strange to me. Before section 6, the authors attempt to provide numerical evidence that the NTK and NNPK kernels are highly close to each other in standard deep learning settings. However, in section 6 they say that the results of dataset distillation of the two kernels could be significantly different (Figure 9). Are these conclusions consistent with one another? I was looking for the authors' explanation for the different results of NNPK and NTK kernels in the paper but could not find such an explanation.

In addition, section 6 seems to suggest that the poor visual quality of the distilled images for the NNPK kernel is a desired property. I cannot understand how the poor quality of distilled images suggests that the NNPK kernel's representation would be useful. On the other hand, I think it could be a sign that NNPK has failed to learn visually relevant features. It seems to me that a desired outcome for a good kernel-based representation will be visually meaningful images that are not revealing about the used samples in support set $X_S$ which seems to be the case in the shown NTK results.

5) The literature review misses several related works on the randomly-weighted neural nets and their connection to kernel functions. Here is a list of such related works that are not discussed in the paper:

1- Daniely et al., "Toward Deeper Understanding of Neural Networks: The Power of Initialization and a Dual View on Expressivity", NeurIPS 2016 \
2- Daniely, " SGD Learns the Conjugate Kernel Class of the Network", NeurIPS 2017 \
3- Yehudai et al, "On the power and limitations of random features for understanding neural networks", NeurIPS 2019, \
4- Shankar et al, "Neural Kernels without Tangents", ICML 2020

---

### Review · Reviewer_4vZZ · 2023-05-25

**Summary Of Contributions:**

In the vein of "neural network gaussian processes" (NNGP) and neural tangent
kernels (NTK), the authors propose a method for obtaining neural kernels
associated to any neural network with logit outputs, which they refer to as the
"neural network prior kernel" (NNPK). The method corresponds to a finite-width
approximation to the NNGP kernel, but with averaging over multiple independent
draws of the random initial network weights. In their experiments, the authors
focus on classification tasks and therein specialize their NNPK construction to
averaging independent draws of a single-logit network in order to obtain their
kernel. They perform comparisons of their approach to NTK approaches, across
many different classification neural network architectures, on datasets ranging
from CIFAR-10 to ImageNet. The purpose of the experiments is largely to
investigate the performance (accuracy) of the NNPK; some connections to
representation learning and privacy are also made.



**Audience:**

Yes

**Broader Impact Concerns:**

None.

**Claims And Evidence:**

No

**Requested Changes:**

- In the methodological discussion, reflect more accurately the fact that the
  proposed NNPK is a variance reduction approach over the typical NNGP
  approach. There should be experiments that investigate this variance
  reduction and its effects in a more controlled fashion across architectures
  (i.e., controlling for width and depth, as well as for the limiting kernel
  being identical), and it would seem to be ideal if the theoretical discussion
  built on prior work in order to attempt to quantify exactly the effect of
  this variance reduction in some simple model architectures, such as deep ReLU
  MLPs (this is, from the cited works above, actually just an application of
  existing results!).
- Additional discussion of prior empirical studies of finite vs. infinite-width
  networks and their kernels, especially since this is one of the core aspects
  of the method (NNPK is just an approach to approximate a kernel defined by an
  integral), especially [7] above.

I believe these changes are critical for acceptance.


**Strengths And Weaknesses:**

- I have some confusion about the fundamental methodological contribution of
  the work. What the authors refer to as the "neural network prior kernel" in
  the equation labeled (NNPK) is identical to the standard NNGP; the difference between
  the authors' approach and the standard approach to working with the NNGP at
  finite width is therefore simply that in equation (1), the typical
  finite-width approximation to the NNGP simply has $n=1$ (i.e., a single
  random draw of the network weights), whereas the authors' NNPK approach
  averages over $n > 1$ draws (possibly, as in (3), over networks with fewer
  than $k$ logits, but from a methodological perspective this seems
  immaterial). This makes it seem like the core distinction of the method
  relative to prior work is to simply introduce averaging over independent
  draws of the stochastic kernel; if this is not something that has been
  proposed in prior work (I am not aware of any, but it seems quite natural so
  I would be surprised if there is no trace of it; I do agree with the authors
  that this is not the same as random features kernel regression), it would
  seem to make sense to me to emphasize this point more clearly than it is
  in the current submission. At present, the authors attempt to distinguish
  their approach from the NNGP along a different line than reflects my
  understanding related here: they mention that NNGP is an infinite-width
  method, whereas their approach is intrinsically finite-width. However, I am
  not sure this accurately reflects existing NNGP methods: it is well-known
  that the finite-width NNGP approximation (i.e., $n=1$ in (1)) converges
  rather quickly in probability [1] (see also nonasymptotic results for
  specific architectures, such as [2]) for networks wider than they are deep;
  the same situation has been studied extensively for other width-depth regimes as well
  [3-5]. Because (1) is simply averaging i.i.d.\ copies of the $n=1$ NNGP
  approximation, the authors' approach is amounting to variance reduction over
  these existing concentration bounds. It therefore seems to me that this is
  the only methodological difference over prior NNGP works, and I think the
  construction of the experiments and the interpretation of their results in
  the submission should reflect this more than the current submission does
  (e.g., the NNPK is framed as a new method in experiments, rather than a
  variance reduction over an existing method; no comparisons are made between
  NNPK and NNGP, rather the authors focus on NNPK vs. NTK even though the NTK
  corresponds to a different limiting kernel, with a larger variance in
  general than the NNGP!).
- Some additional justification of the literature on neural kernels would seem
  to be in order. For example, neural kernels have been explored extensively
  (e.g. [6] and associated works); I am also surprised not to see a discussion
  of the relationship to [7].
- I found the theoretical section of the paper hard to understand. The authors
  seem to be presenting their NNPK as a linearization analogous to the NTK;
  however when one linearizes the network to obtain the NTK, one takes the
  gradient with respect to *all* parameters, whereas for the NNGP (and
  therefore the NNPK), one simply takes the gradient with respect to the last
  layer's weights. Based on the way the method is presented in section 2, I
  would have expected the theoretical section to lead to an equivalence between
  the NNGP and the NNPK, not the NNPK and the NTK.
- In my understanding, the fact that most differences in the empirical results
  between the NTK and the NNPK (after correcting for the fact that the NTK is a
  different limiting kernel than the NNPK, hence should probably be the NNGP)
  arise due to the effects of initialization variance suggests that plots
  should include things like error bars, and it should be more clear in the
  captions/main body how many independent trials are being considered. There
  are parts of the discussion of experiments that allude to this issue (see
  last paragraph before section 7).


[1] https://arxiv.org/abs/2107.01562
[2] https://openreview.net/forum?id=O-6Pm_d_Q-
[3] https://proceedings.neurips.cc/paper/2021/file/412758d043dd247bddea07c7ec558c31-Paper.pdf
[4] http://arxiv.org/abs/2206.02768
[5] http://arxiv.org/abs/2210.00688
[6] http://arxiv.org/abs/2003.02237
[7] http://arxiv.org/abs/2007.15801

---

### Review · Reviewer_vgsT · 2023-05-26

**Summary Of Contributions:**

This work presents Neural Network Prior Kernel (NNPK), which, in comparison with NTK and typical NNGP, could be viewed as a new way to analyze theoretically and empirically certain deep learning problems.

**Audience:**

Yes

**Claims And Evidence:**

No

**Requested Changes:**

Please refer to the concerns mentioned in the above weaknesses.

**Strengths And Weaknesses:**

* Strength

1. The idea of NNPK is interesting and is possible to motivate follow-up works in this direction;

2. The presentation is easy to follow;

* Weaknesses

1. The distinction from NNGP is still not clear to me. More details/demos are suggested to clarify the key difference.

2. How to use the proposed NNPK to study properties of (advanced) DNNs? Authors may need to add more explanation and discussion.

3. The rationality of the assumptions used to support the theoretical results in Section 4 should be further clarified with strong evidence. Otherwise, the reasonableness of the theories may not be guaranteed.

---

### Note · Authors · 2023-06-02

**Comment:**

We would like to thank the reviewers for their time and effort in reviewing our submission. After careful consideration, we have decided to withdraw our submission.  We hope to have the opportunity to present an improved version of our work in the future.

**Withdrawal Confirmation:**

I have read and agree with the venue's withdrawal policy on behalf of myself and my co-authors.